# Pharmacologically Targeting the Fibroblast Growth Factor 14 Interaction Site on the Voltage-Gated Na^+^ Channel 1.6 Enables Isoform-Selective Modulation

**DOI:** 10.3390/ijms222413541

**Published:** 2021-12-17

**Authors:** Nolan M. Dvorak, Cynthia M. Tapia, Aditya K. Singh, Timothy J. Baumgartner, Pingyuan Wang, Haiying Chen, Paul A. Wadsworth, Jia Zhou, Fernanda Laezza

**Affiliations:** Department of Pharmacology and Toxicology, University of Texas Medical Branch, Galveston, TX 75901, USA; nmdvorak@utmb.edu (N.M.D.); cmtapia@utmb.edu (C.M.T.); adsingh@utmb.edu (A.K.S.); tjbaumga@utmb.edu (T.J.B.); wangpingyuan@ouc.edu.cn (P.W.); haichen@utmb.edu (H.C.); pawadswo@utmb.edu (P.A.W.); jizhou@utmb.edu (J.Z.)

**Keywords:** peptidomimetics, protein–protein interactions (PPIs), voltage-gated Na^+^ (Na_v_) channels, fibroblast growth factor 14 (FGF14), medium spiny neurons (MSNs), nucleus accumbens (NAc), neurotherapeutics

## Abstract

Voltage-gated Na^+^ (Na_v_) channels are the primary molecular determinant of the action potential. Among the nine isoforms of the Na_v_ channel α subunit that have been described (Na_v_1.1-Na_v_1.9), Na_v_1.1, Na_v_1.2, and Na_v_1.6 are the primary isoforms expressed in the central nervous system (CNS). Crucially, these three CNS Na_v_ channel isoforms display differential expression across neuronal cell types and diverge with respect to their subcellular distributions. Considering these differences in terms of their localization, the CNS Na_v_ channel isoforms could represent promising targets for the development of targeted neuromodulators. However, current therapeutics that target Na_v_ channels lack selectivity, which results in deleterious side effects due to modulation of off-target Na_v_ channel isoforms. Among the structural components of the Na_v_ channel α subunit that could be pharmacologically targeted to achieve isoform selectivity, the C-terminal domains (CTD) of Na_v_ channels represent promising candidates on account of displaying appreciable amino acid sequence divergence that enables functionally unique protein–protein interactions (PPIs) with Na_v_ channel auxiliary proteins. In medium spiny neurons (MSNs) of the nucleus accumbens (NAc), a critical brain region of the mesocorticolimbic circuit, the PPI between the CTD of the Na_v_1.6 channel and its auxiliary protein fibroblast growth factor 14 (FGF14) is central to the generation of electrical outputs, underscoring its potential value as a site for targeted neuromodulation. Focusing on this PPI, we previously developed a peptidomimetic derived from residues of FGF14 that have an interaction site on the CTD of the Na_v_1.6 channel. In this work, we show that whereas the compound displays dose-dependent effects on the activity of Na_v_1.6 channels in heterologous cells, the compound does not affect Na_v_1.1 or Na_v_1.2 channels at comparable concentrations. In addition, we show that the compound correspondingly modulates the action potential discharge and the transient Na+ of MSNs of the NAc. Overall, these results demonstrate that pharmacologically targeting the FGF14 interaction site on the CTD of the Na_v_1.6 channel is a strategy to achieve isoform-selective modulation, and, more broadly, that sites on the CTDs of Na_v_ channels interacted with by auxiliary proteins could represent candidates for the development of targeted therapeutics.

## 1. Introduction

In excitable cells, voltage-gated Na^+^ (Na_v_) channels enable the initiation and propagation of the action potential [1,2]. Among the nine isoforms of the Na_v_ channel α subunit (Na_v_1.1-Na_v_1.9) that have been described, Na_v_1.1, Na_v_1.2, and Na_v_1.6 are the primary isoforms expressed in the central nervous system (CNS) [1]. In addition to displaying unique electrophysiological profiles, Na_v_1.1, Na_v_1.2, and Na_v_1.6 channels vary with respect to their distributions across neuronal cell types [3,4,5,6,7] and their subcellular distributions [8,9]. Given this heterogeneity of localization, isoform-selective targeting of one of the isoforms could enable targeted neuromodulatory effects. Unfortunately, current therapeutics that target Na_v_ channels lack isoform selectivity due to targeting structural regions of the α subunit that are highly conserved across the nine isoforms, which resultantly confers such drugs with deleterious side-effects due to modulation of off-target Na_v_ channel isoforms [10,11]. As such, the identification of less highly conserved structural regions that are amenable to pharmacological modulation is a necessary prerequisite to fully actualize the potential of Na_v_ channels as targets for neurologic and neuropsychiatric disorders [11].

On account of displaying appreciable amino acid sequence divergence, the C-terminal domains (CTDs) of Na_v_ channels could represent promising sites to pharmacologically target to achieve isoform selective modulation [11,12,13,14,15,16]. In particular, targeting sites on CTDs interacted with by Na_v_ channel auxiliary proteins could represent a novel strategy to achieve isoform-selective modulation given that these are sites that enable endogenously specific intermolecular regulation [11,12,13,17,18,19,20,21,22,23]. In the brain, the Na_v_ channel auxiliary protein fibroblast growth factor 14 (FGF14) interacts with the CTDs of the Na_v_1.1, Na_v_1.2, and Na_v_1.6 channels, and its two splice variants, FGF14-1a and FGF14-1b, differentially regulate the gating and trafficking of the three CNS Nav channel isoforms [12,15,18,24]. Given these functionally unique protein–protein interactions (PPIs) between FGF14 splice variants and CNS Na_v_ channel isoforms, in tandem with FGF14 being an important regulator of neuronal activity and behavior [25,26,27,28,29,30,31,32,33,34], the interaction sites of FGF14 on the CTDs of these Na_v_ channel isoforms could potentially be pharmacologically targeted to develop novel neuromodulators.

Focusing on the FGF14 interaction site on the CTD of the Na_v_1.6 channel on account of the FGF14:Na_v_1.6 complex being central to the generation of electrical outputs of medium spiny neurons (MSNs) of the nucleus accumbens (NAc) [24], which is a critical brain region that regulates reward-related behavior [35], we previously developed a homology model of the PPI interface to guide drug discovery efforts [15]. These investigations identified three clusters of amino acids of FGF14 with interaction sites on the CTD of the Na_v_1.6 channel, namely the Phe-Leu-Pro-Lys (FLPK) and Pro-Leu-Glu-Val (PLEV) motifs on the β12 sheet and the Tyr-Tyr-Val (YYV) motif on the β8/9 loop [15]. In subsequent works, these amino acid sequences were used as scaffolds for the development of peptidomimetics targeting the Na_v_1.6 channel macromolecular complex [24,36,37,38]. Pertinent to the present investigation, we previously presented PW201, also referred to as compound 12, which is derived from the YYV peptide [37]. In our previous study [37], we found that PW201 modulated FGF14:Na_v_1.6 complex assembly, bound appreciably to the CTD of the Na_v_1.6 channel, decreased the Na_v_1.6 channel-mediated transient Na^+^ current (*I*_Na_) in heterologous cells, and had predicted interactions with FGF14′s interaction site on the CTD of Na_v_1.6. In the present investigation, we expand upon these findings and show that whereas PW201 modulates Na_v_1.6 channel-mediated *I*_Na_ in heterologous cells in a dose-dependent manner, the compound displays no effects on Na_v_1.1 channel- or Na_v_1.2 channel-mediated *I*_Na_ at comparable concentrations. Additionally, we show that in MSNs of the NAc, PW201 correspondingly modulates *I*_Na_ and action potential discharge. Overall, these results demonstrate that pharmacologically targeting the FGF14 interaction site on the CTD of the Na_v_1.6 channel enables isoform-selective modulation of the Na_v_1.6 channel and resultantly alters MSN activity, which could collectively represent promising features for the development of future neuromodulators.

## 2. Results

### 2.1. PW201 Has Predicted Interactions with the FGF14^YYV^ Interaction Site on the CTD of the Na_v_1.6 Channel

In our previous study [15], we developed a homology model of the PPI interface between FGF14 and the CTD of the Na_v_1.6 channel using the previously published crystal structure of the CTD of the Na_v_1.5 channel in complex with calmodulin and FGF13 as a template [14]. Through assessment of the homology model, in tandem with biochemical and functional validation modules, we identified the Try158-Tyr159-Val160 motif on the β8/9 loop of FGF14 (FGF14^YYV^) as a “hot segment” [39] at the FGF14:Na_v_1.6 PPI interface [15]. Crucially, these three residues of FGF14 have predicted interaction sites on the CTD of the Na_v_1.6 channel. Based upon FGF14^YYV^ having this predicted interaction site on the CTD of the Na_v_1.6 channel, we first sought to investigate if PW201, which is derived from the YYV motif of FGF14, similarly engaged with residues of the CTD of the Na_v_1.6 channel.

To this end, we employed molecular modeling and docked PW201 with our previously reported homology model of the CTD of Na_v_1.6 [15] (Figure 1A–C). Consistent with PW201’s derivation from the YYV motif of the β8/9 loop of FGF14, the docking study of the compound showed that PW201 docks well with the Na_v_1.6 CTD at the same site where the β8/9 loop of FGF14 interacts. In particular, PW201 forms H-bonds with Asp1833, Met1832, and Arg1891. In addition, the fluorenylmethoxycarbonyl (Fmoc) protecting group added to the N-terminus of the YYV scaffold to improve the compound’s drug-like properties interacts with Arg1866, a residue of the CTD of the Na_v_1.6 channel involved in an intramolecular salt bridge with Asp1846. Collectively considered, these molecular modeling studies provide insights into the putative binding mode of PW201 with the CTD of the Na_v_1.6 channel, which has important implications for understanding its mechanism of action.

### 2.2. PW201 Dose-Dependently Suppresses Na_v_1.6 Channel-Mediated I_Na_ in Heterologous Cells

In our previous study [37], we showed that 20 µM PW201 suppressed Na_v_1.6-mediated *I*_Na_ in heterologous cells, which is an effect similar to that observed due to co-expression of FGF14 with the Na_v_1.6 channel in heterologous cells [12,15,24,38,40,41]. This effect is consistent with the compound’s previously shown direct binding to the CTD of the Na_v_1.6 channel [37]. Additionally, we previously showed that whereas PW201 modulated the peak *I*_Na_ density mediated by Na_v_1.6 channels in heterologous cells, the compound did not affect the voltage dependences of activation or steady-state inactivation [37]. Given these previously shown electrophysiological changes conferred by the compound on Na_v_1.6-mediated *I*_Na_, we sought to assess the dose-dependency of PW201′s effects on Na_v_1.6-mediated peak *I*_Na_ density in HEK293 cells expressing the Na_v_1.6 channel (HEK-Na_v_1.6). To do so, HEK-Na_v_1.6 cells were incubated for 30 min with either vehicle (0.1% DMSO; *n* = 6 cells) or one of seven concentrations of PW201 (range: 1–500 µM; *n* = 4–6 cells per concentration). After incubation, whole-cell patch-clamp electrophysiology was employed, and cells were recorded from using the voltage-clamp protocol shown in Figure 2A. Recordings performed in HEK-Na_v_1.6 cells treated with 0.1% DMSO elicited an average peak current density of −65.6 ± 4.7 pA/pF (*n* = 6 cells). The peak current of each recording was then divided by this average and reported as the percent peak current density (Figure 2B,C).
Figure 2Dose-dependent effects of PW201 on peak *I*_Na_ densities elicited by HEK-Na_v_1.6 cells. (**A**) Representative traces of peak transient currents recorded from HEK-Na_v_1.6 cells treated with 0.1% DMSO (vehicle) or one of seven concentrations of PW201 (range: 1–500 µM). The trace of the peak current from cells treated with 500 µM is not shown to avoid overlap of traces and clarity of representation. Cells were recorded from using the voltage-clamp protocol shown in the inset with a P4 leak cancellation protocol. (**B**) Percentage peak current density plotted as a function of the log concentration of the compound to characterize the dose-dependency of the effects of PW201 on this electrophysiological parameter. Percent peak current was calculated by dividing the peak current density of each recording by the average peak current density of cells treated with 0.1% DMSO. Non-linear regression curve-fitting was performed using GraphPad Prism 8. (**C**) Bar graph representation of individual replicates from dose–response analyses shown in (**B**). (**D**) Comparison of tau of fast inactivation of *I*_Na_ between HEK-Na_v_1.6 cells treated with DMSO and 15 µM PW201. (**E**) Normalized conductance plotted as a function of the voltage to characterize the effects of DMSO (black) and 15 µM PW201 (blue) on the voltage dependence of activation of *I*_Na_ elicited by HEK-Na_v_1.6 cells. Plotted data were fitted with the Boltzmann equation to determine V_1/2_ of activation (see Table 1). (**F**) Normalized current plotted as a function of the voltage to characterize the effects of DMSO (black) and 15 µM PW201 (blue) on the voltage dependence of steady-state inactivation of *I*_Na_ elicited by HEK-Na_v_1.6 cells. Plotted data were fitted with the Boltzmann equation to determine V_1/2_ of steady-state inactivation (see Table 1). Data shown are mean ± SEM. In (**C**,**D**), circles represent individual replicates. In (**C**), significance was assessed using a one-way ANOVA with post hoc Dunnett’s multiple comparisons test. ns, not significant; **, *p* < 0.01; ***, *p* < 0.001; ****, *p* < 0.0001. In (**D**–**F**), significance was assessed using an unpaired *t*-test comparing cells treated with 0.1% DMSO or 15 µM (see Table 1).
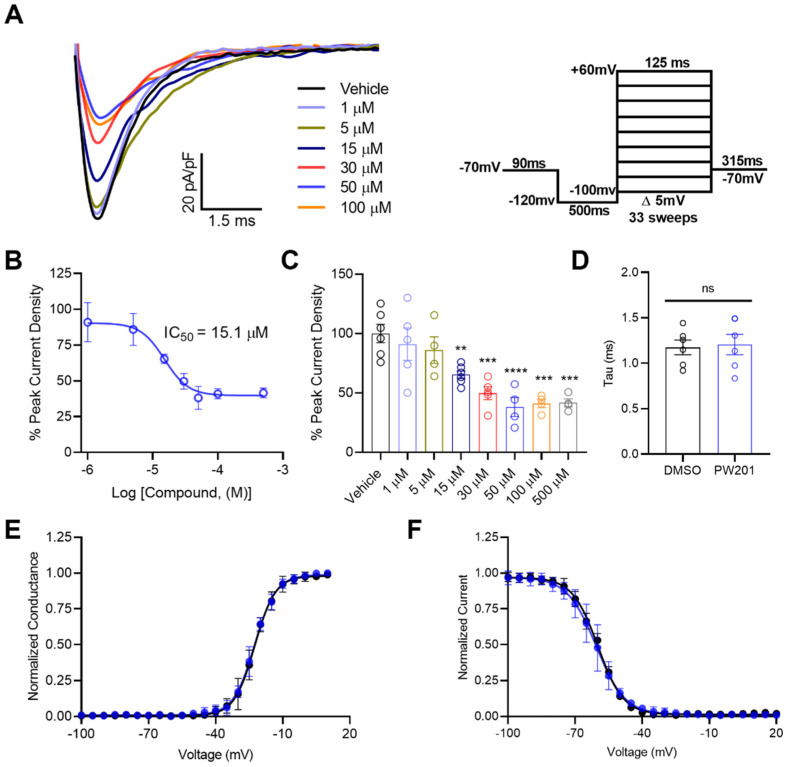

ijms-22-13541-t001_Table 1Table 1Summary of the effects of 15 µM PW201 on Na_v_1.1-, Na_v_1.2-, and Na_v_1.6-mediated currents in heterologous cells ^a^.Na_v_ IsoformConditionPeak Current Density (pA/pF) ^b^Tau of Fast Inactivation (ms) ^c^V_1/2_ of Activation (mV) ^d^V_1/2_ of Steady-State Inactivation (mV) ^e^Na_v_1.1DMSO−139.0 ± 4.8 (8)1.1 ± 0.1 (8)−25.0 ± 1.4 (8)−52.2 ± 1.9 (8)PW201−132.3 ± 3.8 (9)1.0 ± 0.1 (9)−25.3 ± 1.5 (9)−50.7 ± 1.0 (7)Na_v_1.2DMSO−113.4 ± 10.8 (8)1.0 ± 0.1 (8)−25.2 ± 2.5 (8)−54.9 ± 2.3 (6)PW201−107.8 ± 11.9 (7)1.2 ± 0.1 (7)−22.1 ± 1.2 (7)−55.8 ± 3.7 (5)Na_v_1.6DMSO−65.6 ± 4.7 (6)1.2 ± 0.1 (6)−23.1 ± 0.7 (6)−59.7 ± 0.3 (6)PW201−43.4 ± 2.4 (6) ** 1.2 ± 0.1 (6)−21.8 ± 1.3 (6)−61.0 ± 1.3 (6)^a^ Summary of the electrophysiological evaluation of 15 µM PW201 in HEK-Na_v_1.1, HEK-Na_v_1.2, and HEK-Na_v_1.6 cells. Results are expressed as mean ± SEM. The number of independent experiments is shown in parentheses. A Student’s *t*-test comparing cells treated with 0.1% DMSO and 15 µM PW201 was used to determine statistical significance. **, *p* < 0.01. ^b^ Peak current density, which described the number of channels in a conductive (open) state, is a measure of the maximum influx of *I*_Na_ (pA) into the cell normalized to membrane capacitance (pF) to control for variable cell sizes. ^c^ Tau of fast inactivation measures the decay phase of *I*_Na_ to characterize the time required for channels to transition from the conductive (open) state to a nonconductive state resulting from fast inactivation. ^d^ V_1/2_ of activation is a measure of the voltage at which half of the available channels transition from the closed to the conductive (open) state. ^e^ V_1/2_ of steady-state inactivation is a measure of the voltage at which half of channels are available to transition into the conductive (open) state, while the other half are non-conductive due to steady-state (closed-state) inactivation.

In Figure 2B, the half-maximal inhibitory concentration (IC_50_) of PW201 in terms of suppressing Na_v_1.6-mediated peak *I*_Na_ density was determined to be 15.1 µM. This finding is consistent with the bar graph representation of the data in Figure 2C, which shows that statistically significant inhibitory effects on Na_v_1.6-mediated peak *I*_Na_ density are observed at 15 µM but not at single-digit micromolar concentrations. With escalating concentrations, a plateau effect appears to be reached at 50 µM, as the currents from cells treated with 100 and 500 µM PW201 are similarly suppressed. Overall, these dose-dependency studies support the findings of our previous investigation [37] and demonstrate that through direct binding to the CTD of the Na_v_1.6 channel, PW201 is able to dose-dependently affect the peak transient *I*_Na_ density in heterologous cells in a fashion similar to that which is observed due to co-expression of FGF14 with the Na_v_1.6 channel in heterologous cells.

Based on the dose-dependent effects of PW201 on peak *I*_Na_ density observed in Figure 2A–C, we elected to further test the effects of 15 µM PW201 on other electrophysiological parameters, a concentration selected on the basis of it being near the calculated IC_50_ value in Figure 2B. Consistent with our previous investigation, where 20 µM PW201 exerted no effects on the tau of fast inactivation, voltage dependence of activation, or voltage dependence of steady-state inactivation of *I*_Na_ elicited by HEK-Na_v_1.6 cells [37], treatment of HEK-Na_v_1.6 cells with 15 µM PW201 similarly did not affect these parameters (Figure 2D–F). As it pertains to the former, HEK-Na_v_1.6 cells treated with 0.1% DMSO displayed a tau of fast inactivation value of 1.17 ± 0.08 ms, which was not significantly different than the tau of fast inactivation value of HEK-Na_v_1.6 cells treated with 15 µM PW201 (1.21 ± 0.11 ms; *n* = 6 cells per group; *p* = 0.83; Figure 2D). As it relates to the voltage dependence of activation, HEK-Na_v_1.6 cells treated with 0.1% DMSO or 15 µM PW201 displayed V_1/2_ of activation values of −23.1 ± 0.72 mV or −21.8 ± 1.3, respectively (*n* = 6 per group; *p* = 0.41; Figure 2E). Lastly, as it relates to the voltage dependence of steady-state inactivation, HEK-Na_v_1.6 cells treated with 0.1% DMSO or 15 µM PW201 displayed V_1/2_ of steady-state inactivation values of −59.7 ± 0.31 mV or −61.0 ± 1.3 mV, respectively (*n* = 6 per group; *p* = 0.35; Figure 2F). Collectively considered, the results of Figure 2 demonstrate that PW201 confers dose-dependent effects on the *I*_Na_ amplitude, and that the effects of 15 µM PW201 on the *I*_Na_ amplitude are not accompanied by changes in the kinetics or voltage dependences of activation or inactivation of Na_v_1.6 channels.

### 2.3. Profiling the Selectivity of PW201 for the Na_v_1.6 Channel

Having shown previously [37] and demonstrated the dose-dependency (Figure 2) of the effects of PW201 on the transient *I*_Na_ of Na_v_1.6 channels in heterologous cells, we next sought to characterize if these effects of PW201 were selective among CNS Na_v_ channel isoforms. To do so, HEK293 cells stably expressing either Na_v_1.1 (HEK-Na_v_1.1) [24,38,42] or Na_v_1.2 (HEK-Na_v_1.2) [24,38,43] channels were incubated for 30 min with either 0.1% DMSO or 15 µM PW201, a concentration of the ligand selected on the basis of its IC_50_ value for Na_v_1.6 determined in Figure 2B. After incubation, the effects of vehicle and PW201 treatment on Na_v_1.1 and Na_v_1.2 channels were assessed using whole-cell voltage-clamp recordings (Figure 3).

As mentioned above, co-expression of FGF14 with the Na_v_1.6 channel in heterologous cells results in a suppression of Na_v_1.6-mediated *I*_Na_ [12,15,24,38,40,41], similar to the suppression of peak transient *I*_Na_ conferred by treatment of HEK-Na_v_1.6 cells with PW201. Notably, co-expression of FGF14 with the Na_v_1.1 channel [18] and the Na_v_1.2 channel [12] in heterologous cells has similarly been shown to suppress Na_v_1.1- and Na_v_1.2-mediated peak transient *I*_Na_. Despite these conserved modulatory effects of co-expression of FGF14 with Na_v_1.1, Na_v_1.2, and Na_v_1.6 channels on peak *I*_Na_ density in heterologous cells, treatment of only HEK-Na_v_1.6 cells with PW201 results in a suppression of peak *I*_Na_ density, whereas this parameter is unaffected in HEK-Na_v_1.1 (Figure 3A–C) and HEK-Na_v_1.2 (Figure 3I–K) cells treated with 15 µM PW201. Lending further credence to PW201′s isoform-selective effects on the Na_v_1.6 channel, PW201 exerted no modulatory effects on tau of fast inactivation, the voltage dependence of activation, or the voltage dependence of steady-state inactivation of Na_v_1.1 (Figure 3D–H) or Na_v_1.2 (Figure L–P) channels stably expressed in heterologous systems.

### 2.4. PW201 Potentiates the Excitability of MSNs of the NAc through Na_v_ Channel Modulation

Having shown that PW201 modulates Na_v_1.6-mediated *I*_Na_, but not Na_v_1.1- or Na_v_1.2-mediated *I*_Na_, in heterologous cells (Figure 2 and Figure 3, respectively), we next sought to characterize how these collective modulatory effects might alter the activity of MSNs of the NAc. MSNs represent a promising cellular target to be affected by such a ligand as previous studies have shown that FGF14 and Na_v_1.6 channels are enriched in these cells [24]. To test the effects of PW201 on intact MSNs of the NAc, acute brain slice preparations containing the NAc were incubated with either 0.01% DMSO or 15 µM PW201 for 30 min, after which either whole-cell current-clamp (Figure 4A–E) or whole-cell voltage-clamp (Figure 4F–H) electrophysiological recordings were performed.
Figure 4PW201 potentiates the intrinsic excitability and *I*_Na_ of MSNs in the NAc. (**A**) Representative traces of evoked action potentials from MSNs treated with 0.01% DMSO (black) or 15 µM PW201 (blue) in response to increasing current injections (schematic of the current-clamp protocol is shown below representative traces). (**B**) Average number of evoked action potentials at each current step from MSNs treated with 0.01% DMSO (black) or 15 µM PW201 (blue). (**C**) Comparison of the max number of evoked action potentials between MSNs treated with 0.01% DMSO or 15 µM PW201. (**D**) Average instantaneous firing frequencies (IFFs) at each current step from MSNs treated with 0.01% DMSO (black) or 15 µM PW201 (blue). (**E**) Comparison of IFFs of MSNs at the 150 pA current step treated with 0.01% DMSO or 15 µM PW201. (**F**) Representative traces of transient *I*_Na_ of MSNs of the NAc treated with either 0.01% DMSO (black) or 15 µM PW201 (blue) in response to the voltage-clamp protocol shown below the traces. (**G**) Current–voltage relationship for experimental groups descried in (**F**). (**H**) Bar graph derived from (**G**) comparing the peak *I*_Na_ density of MSNs treated with 0.01% DMSO or 15 µM PW201. (**I**) Normalized conductance plotted as a function of the voltage to characterize the effects of 0.01% DMSO (black) or 15 µM PW201 (blue) on the voltage dependence of activation of *I*_Na_ of MSNs. (**J**) Normalized current plotted as a function of the voltage to characterize the effects of 0.01% DMSO (black) or 15 µM PW201 (blue) on the voltage dependence of steady-state inactivation of *I*_Na_ of MSNs. Data are mean ± SEM. In bar graphs, circles represent individual replicates. Significance was assessed using an unpaired *t*-test comparing MSNs treated with 0.1% DMSO and 15 µM PW201. *, *p* < 0.05; **, *p* < 0.01. A table summary of the current-clamp results is shown in Table 2.
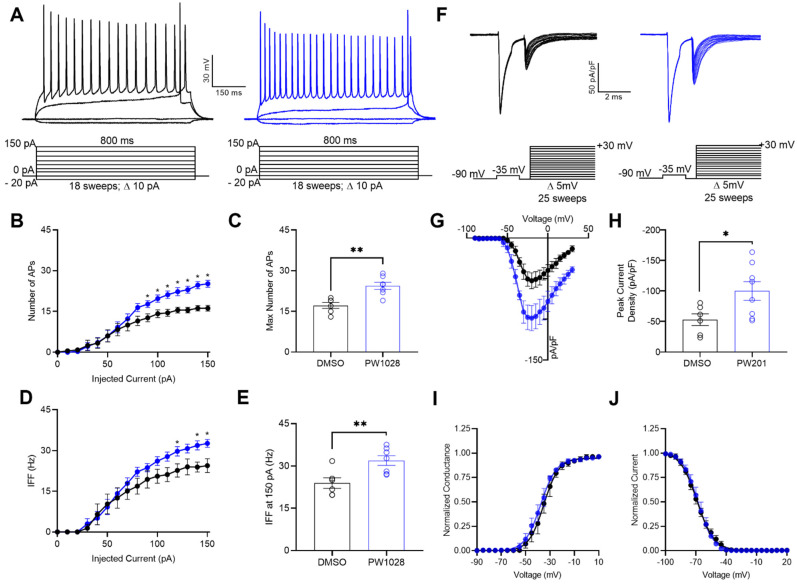

ijms-22-13541-t002_Table 2Table 2Effects of PW201 on passive and electrical properties of MSNs of the NAc ^a^.TreatmentMax Number of APsIFF at 150 pA (Hz)RMP (mV)I_thr_ (pA)V_thr_ (mV)Max Rise (mV/ms)Max Decay (mV/ms)R_in_ (MΩ)Tau (ms)C_m_ (pF)DMSO17.2 ± 1.1 (6)24.5 ± 2.6 (6)−71.7 ± 3.8 (6)40.0 ± 7.3 (6)−43.1 ± 1.4 (6)233.9 ± 33.7 (6)−60.6 ± 4.9 (6)199.6 ± 13.0 (6)25.6 ± 4.6 (6)125.2 ± 14.3 (6) PW20124.4 ± 1.3 (7) **32.7 ± 1.4 (7) **−71.1 ± 1.6 (7)41.4 ± 6.3 (7)−42.9 ± 1.5 (7)227.4 ± 25.1 (7)−58.4 ± 2.8 (7)168.4 ± 13.5 (7)15.6 ± 3.2 (7)91.6 ± 14.1 (7)^a^ Summary of the ex vivo electrophysiological evaluation of PW201 in MSNs of the NAc. Results are expressed as mean ± SEM. The number of independent experiments is shown in parentheses. A Student’s *t*-test comparing MSNs treated with 0.1% DMSO and 15 µM PW201 was used to determine statistical significance. **, *p* < 0.01.

In current-clamp recordings, treatment of MSNs of the NAc with PW201 resulted in a potentiation of their action potential discharge (Figure 4A–C; Table 2). In addition to increasing the maximal firing of MSNs, PW201 also increased the instantaneous firing frequency of these neurons (Figure 4D,E). These effects of PW201 on action potential discharge, coupled with a lack of effects on passive electrical properties, such as resting membrane potential and input resistance (Table 2), suggest that the ligand’s modulatory effects on neuronal excitability likely arise due to changes in Na_v_ channel activity. Such a hypothesis is further supported by the effects of PW201 on Na_v_1.6 channel conductance in heterologous cells (Figure 2), as well as by previous investigations demonstrating changes in the maximal firing and instantaneous firing frequency of neurons due to changes in Nav channel conductance [44,45,46,47].

As an additional test to ensure that the effects on the action potential discharge of MSNs of the NAc conferred by PW201 were mediated by changes in Na_v_ channel activity, whole-cell voltage-clamp recordings of *I*_Na_ were performed in intact MSNs in the acute brain slice preparation using the voltage-clamp protocol shown in Figure 4F. To circumvent space clamp issues that preclude recording of fast gating *I*_Na_ in brain slices using conventional voltage-clamp protocols, we employed a two-pulse step protocol described by Milescu et al. [48] and employed by others [49,50]. This protocol uses a depolarizing pre-pulse step to inactivate Na_v_ channels distant from the recording electrode that is followed shortly afterward with a second step to record Na_v_ channels close to the recording pipette. Using this protocol, we reliably resolved well-clamped *I*_Na_ of intact MSNs in the acute brain slice preparation. MSNs treated with 0.01% DMSO displayed an average peak *I*_Na_ density of −53.0 ± 9.6 pA/pF (*n* = 6), whereas MSNs treated with 15 µM PW201 displayed a significantly increased peak *I*_Na_ density of −100.2 ± 15.2 pA/pF (*n* = 8; *p* < 0.05; Figure 4F–H). This effect provides strong evidence that the compound’s potentiation of action potential discharge of MSNs of the NAc is mediated by changes in the activity of their constituent Na_v_ channels.

In addition to assessing the effects of PW201 on the amplitude of *I*_Na_ of MSNs, the effects of the compound on the voltage dependence of activation (Figure 4I) and the voltage dependence of steady-state inactivation (Figure 4J) of *I*_Na_ of MSNs were also investigated. Consistent with the results shown in Figure 2 demonstrating that 15 µM PW201 modulates the amplitude of Na_v_1.6-mediated *I*_Na_ in heterologous cells without affecting the voltage dependences of activation (Figure 2E) or steady-state inactivation (Figure 2F), treatment of acute brain slice preparations containing the NAc with 15 µM PW201 affected neither the voltage dependence of activation (Figure 4I) nor the voltage dependence of steady-state inactivation (Figure 4J) of the *I*_Na_ of MSNs compared to treatment with 0.01% DMSO. Specifically, MSNs treated with 0.01% DMSO displayed an *I*_Na_ with a V_1/2_ of activation value of −35.4 ± 2.3 mV (*n* = 6), which was not significantly different than MSNs treated with 15 µM PW201 (−38.7 ± 2.0 mV; *n* = 8; *p* = 0.3037; Figure 4I). As it relates to inactivation, the V_1/2_ of steady-state inactivation of *I*_Na_ of MSNs treated with 0.01% DMSO was −69.6 ± 1.5 mV (*n* = 6), which was not significantly different from the V_1/2_ of steady-state inactivation value observed for MSNs treated with 15 µM PW201 (−68.0 ± 2.5 mV; *n* = 8; *p* = 0.6224; Figure 4J). Overall, the results of these recordings performed in MSNs, coupled with the recordings performed in HEK-Na_v_1.6 cells, demonstrate that PW201 affects the *I*_Na_ amplitude without affecting the voltage dependences of activation or steady-state inactivation of *I*_Na_.

## 3. Discussion

PPIs between the pore-forming α subunit of Na_v_ channels and auxiliary proteins regulate channel gating and trafficking [12,17,18,21,22,24,51,52]. Translationally, perturbation of these PPIs gives rise to neural circuity aberrations that are associated with neurologic and neuropsychiatric disorders [33,34], underscoring their role as critical sites for neuromodulation. Despite representing novel pharmacological targets for neuromodulation, such PPIs have historically proven difficult to appreciably modulate using conventional small molecules [53,54,55,56]. As this challenge largely arises from the large size of PPI interfaces making it difficult to identify druggable motifs that could confer functionally relevant modulation of the intermolecular interaction, efforts to map PPI interfaces using chemical probes, such as those employed in the present investigation, are a necessary pre-requisite for the development of small molecule modulators of PPIs.

In our previous work [37], we showed that PW201 modulated FGF14:Na_v_1.6 complex assembly, displayed direct binding to the CTD of the Na_v_1.6 channel, modulated Na_v_1.6-mediated *I*_Na_ in heterologous cells, and docked well with residues that are similarly interacted with by the β8/9 loop of FGF14. In the present work, we expanded upon those findings and showed that whereas PW201 modulated Na_v_1.6-mediated *I*_Na_ in heterologous cells in a dose-dependent manner with an IC_50_ of 15 µM (Figure 2), the ligand displayed no effects on Na_v_1.1 or Na_v_1.2 channels in heterologous cells when similarly tested at 15 µM (Figure 3). These findings could suggest that the ligand binding site of PW201 on the CTD of the Na_v_1.6 channel is not conserved among the Na_v_1.1 or Na_v_1.2 channels; however, extensive structural and biophysical studies are required to unequivocally substantiate such a hypothesis. Nevertheless, these findings, coupled with the molecular modeling of PW201 shown in Figure 1, could help guide future rational design efforts seeking to develop isoform-selective small molecule modulators of the Na_v_1.6 channel.

In addition to demonstrating isoform-selective effects of PW201 on Na_v_1.6 channels in heterologous cells, we also assessed the effects of PW201 on the *I*_Na_ and intrinsic excitability of MSNs of the NAc. MSNs represent the principal cell type of the NAc [57,58], are highly vulnerable to neurodegeneration [59], and are enriched with FGF14 and the Na_v_1.6 channel [24]. In current-clamp and voltage-clamp recordings, PW201 was shown to potentiate the action potential discharge (Figure 4A–E) and increase the *I*_Na_ amplitude of MSNs of the NAc (Figure 4F–H), respectively. Importantly, changes in Na_v_ channel conductance, such as those conferred by PW201, have previously been shown to increase neuronal excitability and confer changes in the instantaneous firing frequencies of neurons [44,45,46,47]. Coupled with the findings observed in heterologous cells, these results demonstrate that pharmacological manipulation of the Na_v_1.6 channel achieved through targeting its PPI site with FGF14 can alter the activity of cells in clinically relevant brain regions, underscoring the potential translational value of the target for neurologic and neuropsychiatric diseases.

One seemingly paradoxical effect of PW201 is that whereas the ligand suppresses Na_v_1.6-mediated *I*_Na_ in heterologous cells, the compound increases the *I*_Na_ of MSNs in the acute brain slice preparation. However, the opposite effects of FGF14, the protein from which PW201 is derived, in heterologous cells versus neurons is widely recognized [12,17,24]. Specifically, co-expression of FGF14 with the Na_v_1.6 channel in heterologous systems has previously been shown to suppress Na_v_1.6-mediated *I*_Na_ [12,15,24,38,40,41,60,61], whereas over-expression of FGF14 in neurons has been shown to increase *I*_Na_ [17]. As such, these opposite effects observed for PW201 in heterologous cells versus in neurons are unsurprising and provide supporting evidence for the compound functioning as a partial pharmacological mimic of FGF14.

Although PW201 is anticipated to not be blood–brain barrier permeable due to its high molecular weight and total polar surface area, the findings of the present investigation will inform rational design efforts to develop isoform-selective small molecule modulators of the Na_v_1.6 channel macromolecular complex. Such neuromodulators that exert their effects through targeting of PPI interfaces within the CNS will represent an entirely novel class of neurotherapeutics and will demonstrate that PPIs represent hundreds of viable and unexplored targets for CNS drug development.

## 4. Materials and Methods

### 4.1. Molecular Docking

The molecular docking study was performed using Schrödinger Small-Molecule Drug Discovery Suite (Schrödinger, LLC, New York, NY, USA). The FGF14:Na_v_1.6 homology model was built using the FGF13:Na_v_1.5:CaM ternary complex crystal structure (PDB code: 4DCK) as a template [14]. The FGF14:Na_v_1.6 CTD homology model was prepared with Schrödinger Protein Preparation Wizard using default settings. The SiteMap (Schrödinger, LLC) calculation was performed, and a potential binding site was identified on the PPI interface of FGF14 and the CTD of the Na_v_1.6 channel. The docking was performed on the CTD of Na_v_1.6 after removing the FGF14 chain structure. The grid center was chosen on the Na_v_1.6 CTD at the previously identified binding site with a grid box sized in 24 Å covering the PPI surface on the Na_v_1.6 CTD. The 3D structure of PW201 was created using Schrödinger Maestro and a low-energy conformation was generated using LigPrep. Docking was then employed with Glide using the SP precision. Docked poses were incorporated into Schrödinger Maestro for a ligand–receptor interactions visualization. The top docked pose of PW201 was superimposed with the FGF14:Na_v_1.6 CTD complex homology model for an overlay analysis.

### 4.2. Chemicals

The synthetic route, as well as the chemical properties, of PW201 were previously described [37]. Lyophilized PW201 powder (purity > 95%) was reconstituted in 100% dimethyl sulfoxide (DMSO; Sigma-Aldrich, St. Louis, MO, USA) to achieve stock concentrations of 50 mM, which were frozen and stored at −20 °C until being thawed and further diluted for experimental purposes.

### 4.3. Cell Culture

HEK293 cells were maintained in a 1:1 mixture of Dulbecco’s Modified Eagle Medium (DMEM) with 1 g/L glucose and F-12 (Invitrogen, Carlsbad, CA, USA) that was further supplemented with 10% fetal bovine serum, 100 units/mL of penicillin, and 100 µg/mL streptomycin (Invitrogen). Cells were maintained at 37 °C. The HEK293 cells stably expressing hNa_v_1.1 [42], hNa_v_1.2 [43], and hNa_v_1.6 [15,24,40,41] channels have previously been described. These cells were maintained according to general cell culture protocols, with the caveat that 500 µg/mL of G418 (Invitrogen) was used to maintain stable hNa_v_1.2 and hNa_v_1.6 expression and 80 µg/mL of G418 was used to maintain stable expression of hNa_v_1.1.

### 4.4. Animals

C57/BL6J mice were purchased from Jackson Laboratory (Bar Harbor, ME, USA). Mice were housed in the University of Texas Medical Branch vivarium, which operates in compliance with the United States Department of Agriculture Animal Welfare Act, the NIH Guide for the Care and Use of Laboratory Animals, the American Association for Laboratory Animal Science, and Institutional Animal Care and Use Committee approved protocols.

### 4.5. Electrophysiology

#### 4.5.1. General

Borosilicate glass pipettes (Harvard Apparatus, Holliston, MA, USA) with resistance of 1.5–3 MΩ were fabricated using a PC-100 vertical Micropipette Puller (Narishige International Inc., East Meadow, NY, USA). Recordings were obtained using an Axopatch 200B amplifier (Molecular Devices, Sunnyvale, CA, USA). Membrane capacitance and series resistance were estimated using the dial settings on the amplifier, and capacitive transients and series resistances were compensated by 70–80%. Data acquisition and filtering occurred at 20 and 5 kHz, respectively, before digitization and storage. Clampex 9 software (Molecular Devices) was used to set experimental parameters, and electrophysiological equipment was interfaced to this software using a Digidata 1200 analog–digital interface (Molecular Devices). Analysis of electrophysiological data was performed using Clampfit 11 software (Molecular Devices) and GraphPad Prism 8 software (La Jolla, CA, USA). Results were expressed as mean ± standard error of the mean (SEM). Except where otherwise noted, statistical significance was determined using a Student’s *t*-test comparing cells treated with vehicle (DMSO) or PW201, with *p* < 0.05 being considered statistically significant.

#### 4.5.2. Whole-Cell Voltage-Clamp Recordings

Whole-cell voltage-clamp recordings in heterologous cell systems were performed as previously described [37,38]. Briefly, cells cultured as described in Section 4.3 were dissociated using TrypLE (Gibco, Waltham, MA, USA) and re-plated onto glass cover slips. After allowing cells at least 2–3 h to adhere, cover slips were transferred to a recording chamber. The recording chamber was filled with an extracellular recording solution comprised of the following salts: 140 mM NaCl; 3 mM KCl; 1 mM MgCl_2_; 1 mM CaCl_2_; 10 mM HEPES; and 10 mM glucose (final pH = 7.3; all salts purchased from Sigma-Aldrich, St. Louis, MO, USA). For control recordings, DMSO was added to the extracellular solution to reach a final concentration of 0.1%. For recordings to characterize the effects of PW201, the compound was added to the extracellular solution to reach the desired final concentration. Cover slips were incubated for 30 min in either vehicle only or PW201 containing extracellular solutions prior to the start of recordings. For voltage-clamp recordings, recording pipettes were filled with an intracellular solution comprised of the following salts: 130 mM CH_3_O_3_SCs; 1 mM EGTA; 10 mM NaCl; and 10 mM HEPES (pH = 7.3; all salts purchased from Sigma-Aldrich). After GΩ seal formation and entry into the whole-cell configuration, two voltage-clamp protocols were employed. The current-voltage (IV) protocol entailed voltage steps from −100 to +60 mV from a holding potential of −70 mV. The voltage dependence of steady-state inactivation protocol entailed a paired-pulse protocol during which, from the holding potential, cells were stepped to varying test potentials between −100 mV and +20 mV prior to a test pulse to −20 mV.

#### 4.5.3. Voltage-Clamp Data Analysis

Current densities were obtained by dividing the Na^+^ current (*I*_Na_) amplitude by the membrane capacitance (C_m_). Current–voltage relationships were then assessed by plotting the current density as a function of the applied voltage. Tau of fast inactivation was calculated by fitting the decay phase of currents at the −10 mV voltage step with a one-term exponential function. To assess the voltage dependence of activation, conductance (G_Na_) was first calculated using the following equation:$$G_{\mathrm{Na}}=\frac{I_{\mathrm{Na}}}{\left(V_{m}-E_{\mathrm{rev}}\right)}$$
where *I*_Na_ is the current amplitude at voltage V_m_, and E_rev_ is the Na^+^ reversal potential. Activation curves were then generated by plotting normalized G_Na_ as a function of the test potential. Data were then fitted with the Boltzmann equation to determine V_1/2_ of activation using the following equation:$$\frac{G_{\mathrm{Na}}}{G_{\mathrm{Na},\;\max}}=1+e^{V_{a}-E_{m}/k}$$
where G_Na,max_ is the maximum conductance, V_a_ is the membrane potential of half-maximal activation, E_m_ is the membrane voltage, and k is the slope factor. For steady-state inactivation, the normalized current amplitude (*I*_Na_/*I*_Na,max_) at the test potential was plotted as a function of the pre-pulse potential (V_m_) and fitted using the Boltzmann equation:$$\frac{I_{\mathrm{Na}}}{I_{\mathrm{Na},\max}}=\frac{1}{1+e^{V_{h}-E_{m}/k}}$$
where V_h_ is the potential of half-maximal inactivation, E_m_ is the membrane voltage, and k is the slope factor.

#### 4.5.4. Acute Brain Slice Preparation

Whole-cell current-clamp and whole-cell voltage-clamp recordings were performed in acutely pre-prepared coronal brain slices containing the NAc from mice described in Section 4.4 that were 33–50 days old. For brain slice preparation, mice were anesthetized using isoflurane (Baxter, Deerfield, IL, USA) and quickly decapitated. After decapitation, brains were dissected and 300 µM coronal slices containing the NAc were prepared with a vibratome (Leica Biosystems, Buffalo Grove, IL, USA) in a continuously oxygenated (mixture of 95% O_2_/5% CO_2_) and chilled tris-based artificial cerebrospinal fluid (aCSF) containing the following salts: 72 mM Tris-HCL; 18 mM Tris-Base; 1.2 mM NaH_2_PO_4_; 2.5 mM KCl; 20 mM HEPES; 20 mM sucrose; 25 mM NaHCO_3_; 25 mM glucose; 10 mM MgSO_4_; 3 mM Na pyruvate; 5 mM Na ascorbate; and 0.5 mM CaCl_2_ (pH = 7.4 and osmolarity = 300–310 mOsm; all salts purchased from Sigma-Aldrich). Prepared slices were first transferred to a continuously oxygenated and 31 °C recovery chamber containing fresh tris-based aCSF for 15 min. After 15 min, slices were transferred to a continuously oxygenated and 31 °C chamber containing standard aCSF, which was comprised of the following salts: 123.9 mM NaCl; 3.1 mM KCl; 10 mM glucose; 1 mM MgCl_2_; 2 mM CaCl_2_; 24 mM NaHCO_3_; and 1.16 mM NaH_2_PO_4_ (pH = 7.4 and osmolarity = 300–310 mOsm; all salts were purchased from Sigma-Aldrich). After at least 30 min of recovery in standard aCSF, slices were incubated for 30 min in a chamber containing continuously oxygenated and 31 °C standard aCSF treated with either 0.01% DMSO or 15 µM PW201 before recording.

#### 4.5.5. Whole-Cell Current-Clamp Recordings

After incubating for 30 min in either 0.01% DMSO or 15 µM PW201, slices were transferred to a recording chamber perfused with continuously oxygenated and heated standard aCSF. Somatic recordings of MSNs were then performed using electrodes filled with an internal solution comprised of the following salts: 145 mM K-gluconate; 2 mM MgCl_2_; 0.1 mM EGTA; 2.5 mM Na_2_ATP; 0.25 mM Na_2_GTP; 5 mM phosphocreatine; and 10 mM HEPES (pH = 7.2 and osmolarity = 290 mOsm; all salts were purchased from Sigma-Aldrich). After GΩ formation and entry into the whole-cell configuration, the amplifier was switched to *I* = 0 mode for approximately 1 min to determine the resting membrane potential before switching to current-clamp mode to assess intrinsic excitability. During this 1 min interval in *I* = 0 mode, the following cocktail of synaptic blockers was perfused to halt changes in excitability driven by synaptic activity: 20 µM bicuculine; 20 µM NBQX; and 100 µM AP5 (synaptic blockers purchased from Tocris, Bristol, UK). To assess intrinsic excitability, evoked APs were measured in response to a range of current injections from −20 to +150 pA. Current steps were 800 ms in duration, and the change in the injected current between steps was 10 pA.

#### 4.5.6. Current-Clamp Data Analysis

The maximum number of APs was determined by quantifying the maximum number of APs an MSN fired at any current step during the evoked protocol. The average instantaneous firing frequency was determined by calculating the mean value of the instantaneous firing frequency between APs at a given current step. The current threshold (I_thr_) was defined as the current step at which at least one AP was evoked. Voltage threshold (V_thr_) was defined as the voltage at which the first-order derivative of the rising phase of the AP exceeded 10 mV/ms [62]. The maximum rise and maximum decay of APs were defined as the maximal derivative value (dV/dt) of the depolarizing and repolarizing phases of the AP, respectively [63].

#### 4.5.7. Ex Vivo Whole-Cell Voltage-Clamp Recordings of I_Na_

The extracellular solution used for current-clamp recordings was also used to record *I*_Na_ of MSNs ex vivo, with the caveat that the superfusing solution was supplemented with 120 µM CdCl_2_ (Sigma-Aldrich) to block Ca^2+^ currents. The intracellular solution to record *I*_Na_ of MSN ex vivo contained the following salts (in mM): 100 mM Cs-gluconate (Hello Bio Inc., Princeton, NJ, USA); 10 mM tetraethylammonium chloride; 5 mM 4-aminopyridine; 10 mM EGTA; 1 mM CaCl_2_; 10 mM HEPES; 4 mM Mg-ATP; 0.3 mM Na_3_-GTP; 4 mM Na_2_-phosphocreatine; and 4 mM NaCl (pH = 7.4 and osmolarity = 285 ± 5 mOsm/L; CsOH used to adjust pH and osmolarity; all salts except Cs-gluconate purchased from Sigma-Aldrich). After GΩ formation and entry into the whole-cell configuration, the same cocktail of synaptic blockers as used for the current-clamp recordings was perfused to block synaptic currents. Transient *I*_Na_ was elicited using the voltage-clamp protocol shown in Figure 4F and as described elsewhere [48,49,50]. *I*_Na_ density was calculated by normalizing the *I*_Na_ response by C_m_.

## Figures and Tables

**Figure 1 ijms-22-13541-f001:**
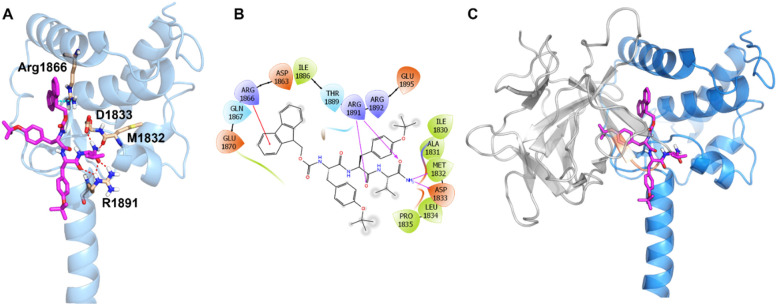
Molecular modeling of PW201 in complex with the homology model of the CTD of the Na_v_1.6 channel (modified from Dvorak et al., 2020. *Molecules*. PMID: 32722255). (**A**) Ribbon representation of PW201 (magenta) docked with the homology model of the CTD of Na_v_1.6 (blue). Key interaction residues are shown as sticks, H-bonds are shown as red dashed lines, and Pi–cation interactions are shown as cyan dashed lines. (**B**) Interaction diagram of PW201′s predicted binding site with the homology model of the CTD of Na_v_1.6. H-bonds are shown as purple lines and Pi–cation interactions are shown as red lines. (**C**) Docked pose of PW201 with the homology model of the CTD of the Na_v_1.6 channel overlaid with FGF14. The Na_v_1.6 CTD is shown as blue ribbon and FGF14 as gray ribbon. Ligand is shown as a magenta stick and the YYV motif of the FGF14 β8/9 loop is highlighted in orange.

**Figure 3 ijms-22-13541-f003:**
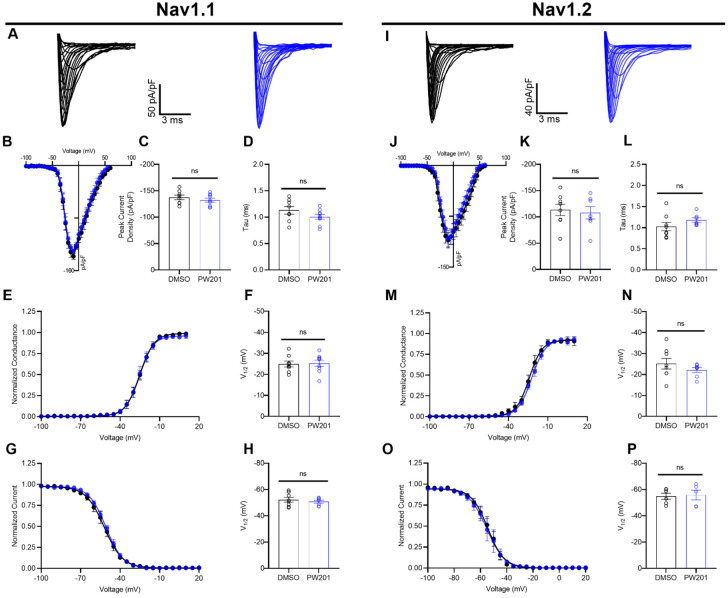
Electrophysiological evaluation of the effects of PW201 on Na_v_1.1 and Na_v_1.2 channels. (**A**,**I**) Representative traces of transient Na^+^ currents elicited by the indicated cell type treated with either 0.1% DMSO (black) or 15 µM PW201 (blue). (**B**,**J**) Current–voltage relationships for cells of the indicated type treated with either 0.1% DMSO (black) or 15 µM PW201 (blue). (**C**,**K**) Comparison of the peak current density for the experimental groups described in (**B**) and (**J**), respectively. (**D**,**L**) Comparison of tau of fast inactivation of HEK-Na_v_1.1 and HEK-Na_v_1.2 cells, respectively, treated with either 0.1% DMSO (black) or 15 µM PW201 (blue). (**E**,**M**) Normalized conductance plotted as a function of the voltage for HEK-Na_v_1.1 and HEK-Na_v_1.2 cells, respectively, that were treated with 0.1% DMSO (black) or 15 µM PW201 (blue) to characterize the effects of vehicle and compound treatment on the voltage dependencies of activation of Na_v_1.1 and Na_v_1.2 channels. (**F**,**N**) Comparison of V_1/2_ of activation of transient Na^+^ currents elicited by HEK-Na_v_1.1 and HEK-Na_v_1.2 cells, respectively, that were treated with 0.1% DMSO or 15 µM PW201. (**G**,**O**) Normalized current plotted as a function of the voltage for HEK-Na_v_1.1 and HEK-Na_v_1.2 cells, respectively, that were treated with 0.1% DMSO (black) or 15 µM PW201 (blue) to characterize the effects of vehicle and compound treatment on the voltage dependencies of steady-state inactivation of Na_v_1.1 and Na_v_1.2 channels. (**H**,**P**) Comparison of V_1/2_ of steady-state inactivation of transient Na^+^ currents elicited by HEK-Na_v_1.1 and HEK-Na_v_1.2 cells, respectively, that were treated with 0.1% DMSO (black) or 15 µM PW201 (blue). Data are mean ± SEM. In bar graphs, circles represent individual replicates. Significance was assessed using an unpaired *t*-test comparing cells treated with 0.1% DMSO and 15 µM PW201. ns, not significant. A table summary of results is shown in Table 1.

## Data Availability

Data included in this study are available upon request from the corresponding author.

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
