# Peer review of "Pharmacologically Targeting the Fibroblast Growth Factor 14 Interaction Site on the Voltage-Gated Na+ Channel 1.6 Enables Isoform-Selective Modulation"

_ijms, 2021, doi:10.3390/ijms222413541_

Round 1

Reviewer 1 Report

The submitted manuscript is very well organized and the scientific information is very clear.

The research is a development of earlier achievements. The results may be interesting for neurotherapy specialists as well as for scientists looking for drugs targeting the CNS.

My opinion is very positive.

Author Response

We thank this reviewer for the appreciation of our work. We are pleased to read this enthusiastic review of our work that endorses the manuscript for publication in the form in which it was originally submitted. 

Reviewer 2 Report

The manuscript entitled “Pharmacological targeting the FGF14 interaction site…” is a continuation of a previous work “Bi-directional modulation…”, published in Molecules, 2020 by the same group in which they described peptides (peptidomimetics) capable of modulating FGF14: Nav1.6. PW201 induced negative and positive allosteric modulation of the FGF14: Nav1.6 complex assembly. In the submitted manuscript, the authors focused on the FGF14 and CTD interactions of the Nav1.6 channel. The results described in this manuscript show that specific targeting of the site of FGF14 interaction on CTD Nav1.6 enables isoform selective channel modulation to be achieved. This approach provide opportunity to find suitable candidates for the development of targeted therapeutic agents against MSN dysfunction. The paper sounds very scientific. Presented experiments are done carefully and the conclusions obtained are supported by the data shown. In turn, the data is presented in a readable way. Everything is written in easy way to get through it.

In my opinion, this manuscript is suitable for publication in the IJMS already in its current form.

Author Response

(The authors gave the same response as above.)

Reviewer 3 Report

This is a very interesting and valuable article. It should br accepted after a minor revision.

I would suggest that the Authors should supply the data shown in Figure 2 by results of studies on the influence of the PW201 at one chosen concentration (e.g. 15 mikrom) on the steady-state activation, steady-state inactivation and the inactivation time constant of Nav1.6 currents, such as they did in the case of Nav1.1 and Nav1.2 currents (Fig. 3). The calculated values of peak current intensity, tau of inactivation, V1/2 of activtion and inactivation of Nav1.6 channels upon application of DMSO and PW201 should be added into the Table 1 and compared to the values calculated for Nav1.1 and Nav1.2 currents. The same study should be performed in case of Nav1.6 currents recorded in slices (Fig. 4F-H). This is of importance because it seems likely that application of PW201 decreases the inactivation rate, shifts the steady-state inactivation curve towards more positive membrane potentials and probably also increases the rate of recovery from inactivation. Slow- down of inactivation may be responsible for increase of peak current observed in Fig. 4F, whereas the shift of steady-state inactivation curve towards posititive membrane potentials combined with acceleration of recovery from inactivation may be responsible for incerease of intrinsic excitability of MSNs (Fig. 4A-E).

Minor point: the Authors should explain all abbreviations in the Table 2.

Author Response

1) This is a very interesting and valuable article. It should be accepted after a minor revision.

Response: We thank this reviewer for the appreciation of our work.

2) I would suggest that the Authors should supply the data shown in Figure 2 by results of studies on the influence of the PW201 at one chosen concentration (e.g. 15 mikrom) on the steady-state activation, steady-state inactivation and the inactivation time constant of Nav1.6 currents, such as they did in the case of Nav1.1 and Nav1.2 currents (Fig. 3).

Response: We thank this reviewer for the feedback regarding the presentation of the electrophysiology data collected in heterologous cells. The additional data requested is shown in Figure 2D-F of the revised version of the manuscript.

3) The calculated values of peak current intensity, tau of inactivation, V1/2 of activtion and inactivation of Nav1.6 channels upon application of DMSO and PW201 should be added into the Table 1 and compared to the values calculated for Nav1.1 and Nav1.2 currents.

 Response: We thank this reviewer for the feedback regarding the presentation of the electrophysiology data collected in heterologous cells. The requested information is now included in Table 1.

4) The same study should be performed in case of Nav1.6 currents recorded in slices (Fig. 4F-H). This is of importance because it seems likely that application of PW201 decreases the inactivation rate, shifts the steady-state inactivation curve towards more positive membrane potentials and probably also increases the rate of recovery from inactivation. Slow- down of inactivation may be responsible for increase of peak current observed in Fig. 4F, whereas the shift of steady-state inactivation curve towards posititive membrane potentials combined with acceleration of recovery from inactivation may be responsible for incerease of intrinsic excitability of MSNs (Fig. 4A-E).

Response: We thank this reviewer regarding the voltage-clamp studies done in slices. The additional data requested is shown in Figure 4I,J of the revised version of the manuscript.

5) Minor point: the Authors should explain all abbreviations in the Table 2.

Response: We thank this reviewer regarding the terminology and abbreviations employed throughout the manuscript. The abbreviation and terminology employed for voltage-clamp studies is described in the legend of Table 1 of the revised version of the manuscript. A description of the abbreviations and terminology employed in Table 2 is included in the Methods section 4.5.6 titled “Current-clamp data analysis.”